# Response of Care Services for Patients with HIV/AIDS during a Pandemic: Perspectives of Health Staff in Bolivia

**DOI:** 10.3390/ijerph192013515

**Published:** 2022-10-19

**Authors:** Liseth Lourdes Arias López, Maria Teresa Solis-Soto

**Affiliations:** 1OH TARGET Competence Center, Universidad San Francisco Xavier de Chuquisaca, Estudiantes, 96, Sucre P.O. Box 212, Bolivia; 2Occupational and Environmental Epidemiology & Net Teaching Unit, Institute for Occupational, Social and Environmental Medicine, University Hospital Munich (LMU), Ziemssenstr. 1, 80336 Munich, Germany; 3Center for International Health, University Hospital Munich (LMU), Ziemssenstr. 1, 80336 Munich, Germany

**Keywords:** HIV, health response, COVID-19, health determinants

## Abstract

The COVID-19 pandemic has caused an unprecedented crisis striking health services, generating risks of setbacks in health care and affecting the most vulnerable populations such as HIV patients. This study aims to explore the impact of the COVID-19 pandemic on the operational management of health services for people living with HIV/AIDS in Cochabamba, Bolivia. We applied a qualitative approach using semi-structured in-depth interviews with ten key health professionals who care for patients with HIV/AIDS in Cochabamba, Bolivia. Interviews were transcribed verbatim and uploaded to Atlas.ti software for analysis. We used an ethnographic approach within the interpretive paradigm to carry out the thematic analysis, considering the “Determinants of health systems resilience framework” of five dimensions developed by the World Health Organization. Even though the provision of services in public care services was not interrupted during the COVID-19 pandemic, health service delivery was severely affected. Digital technology could be used to compensate in urban areas. Regarding the distribution of medications, adaptative strategies to reduce patient complications were implemented. Unfortunately, the complementary tests availability was limited. The COVID-19 pandemic had a significant impact on HIV/AIDS patient care services in Cochabamba, with repercussions for HIV treatment access and virologic suppression.

## 1. Introduction

The pandemic caused by the new coronavirus (COVID-19) has affected the organization and functioning of health care systems globally [1]. It has challenged local, national, and global capacities to prepare and respond to emergencies [2]. Countries have reorganized health services [1] and have had to take measures to reduce contagions, such as isolation and social distancing [3], which in turn caused economic and social disruption [2]. It also affected the mental health of the population and quality of life, reporting feelings of fear and apprehension [4]. In Bolivia, and other countries in Latin America, the pandemic has affected health systems and exacerbated existing problems. It has generated a so called “health debt” caused by the postponement of care and treatment to prioritize the control of the pandemic. On the other hand, health personnel reported the need for emotional and economical support, conflicts with the relatives of COVID infected patients, and changes in their usual work functions, which have also affected the regular provision of services [5].

This reorganization has generated risks of setbacks in health [1], affecting the most vulnerable populations [2], mainly individuals with chronic illness [6] and people with compromised immune systems [7], including people living with HIV/AIDS (PLWHA). In 2020, Bolivia reported 27,930 cases of HIV/AIDS, 19.3% of them in Cochabamba, with 5310 accumulated cases [8]; even though the prevalence is low compared to other countries, it has been increasing, especially among people aged 15 to 49 representing 0.15% of the total population [9]. This age group was probably the most affected by the disruption of HIV epidemic control [10].

The COVID-19 pandemic decreased access to HIV testing and prevention and worsened HIV treatment access and virology suppression [10]. A study in 28 countries by UNAIDS showed disruptions in access to antiretroviral (ARV) treatment as well as a significant decline in HIV testing and antiretroviral therapy (ART) initiation [11]. Additionally, a survey conducted among African countries showed that 19 of 33 (58%) countries experienced disruptions in ARV treatment [11].

A study in Ethiopia showed a significant decrease in HIV testing and detection along with enrollment in ART [12]. Another study in Guatemala found a reduction in testings of 54.7%; and also found that deaths from opportunistic infections at 90 days were 10.7% higher in 2020 compared with 2019. A study conducted in Argentina on 1336 PLWHA reported reduction in medication adherence (33%) during the lockdown [6].

This interruption of access to ART could increase mortality by up to 10% among PLWHA in low and middle-income countries [10], and has the potential risk of contributing to HIV drug resistance [11]. The most critical determinant of HIV-related mortality was reported to be the interruption of ART supply. It is estimated that the number of additional deaths caused by an interruption of three months of ART supply to 40% of individuals is similar to the number of lives saved from COVID-19 through social distancing [13]. Although there aren’t many studies in Latin America, a study in Brazil showed that PLWHA had adequate adherence to antiretrovirals. Regardless, all were in social isolation, without follow-up appointments, with access to the health service only to receive antiretrovirals [14].

Other services were also disrupted: testing, viral load monitoring, prevention services, pre-exposure prophylaxis, and early infant diagnosis to prevent mother-to-child transmission [11]. A study in Boston reported a 72% decrease in pre-exposure Prophylaxis initiations during the pandemic, although this can perhaps be explained by a change in sexual practices [10]. In Brazil a qualitative study that explored barriers to access HIV post-exposure prophylaxis perceived by users and professionals, described the lack of knowledge about prophylaxis, centralization of healthcare services and stigmas that permeate the structures of healthcare services as main barriers [15].

The lack of access to regular health services for these patients came about for several reasons: Many HIV/AIDS prevention and control centers worldwide have been converted into COVID-19 treatment centers. Public transportation restrictions also limited access to health care services [7]. This situation and the perceived fear of contracting COVID-19 have made this group’s situation more vulnerable [16]. Also, HIV care resources, including healthcare personnel, have been channeled into curbing the COVID-19 pandemic [16]. Many people also experienced reduced income and social isolation [6], affecting HIV susceptibility and risk [10]. The PLWHA are vulnerable to both the direct and indirect consequences of the pandemic [17].

Although some international experiences have been reported to reinforce the HIV/AIDS services to guarantee ARVs distribution during the pandemic, there are still uncertainties about the situation of assistance to PLWHA in countries where the economy was highly affected [16], such as in Bolivia. On the other hand, while the success assessment of these experiences has mainly used COVID-19 mortality and morbidity indicators, it is still needed to analyze “collateral effects” such as set back access for care and treatment among patients with chronic illnesses [3] considering all levels of the health systems [18].

Chronic illnesses complexity for care and treatment (such as HIV/AIDS), and the threat of new emerging diseases [2], added to the increased demand of face to face health services [19], even in health crisis contexts (such as the COVID pandemic), make it necessary to strengthen the resilience of health systems, countries’ preparedness, and response strategies considering their socio-economic and cultural context [20]. Collaborative multidisciplinary policy approaches, including stakeholders from all relevant sectors, should be employed [18]. The framework determinants of health systems resilience, developed by the WHO, puts forward the following dimensions which are relevant to our study:(a)Community engagement: this is the core of this framework. It includes strategies such as building partnerships with local leaders and working alongside community members to tailor messages and campaigns.(b)Governance, finance, and collaboration across the sector: this includes government decisions related to healthcare infrastructures, regulations, and guidelines, defining access to medication and treatment, the provision of health coverage, and financing [2].(c)Health service delivery: associated with more access barriers in low and middle-income countries [1], such as Bolivia. The loss of social services and in-person support in HIV clinics affected the health of PLWHA. It is consistent with a study on PLWHA homelessness in an HIV clinic in San Francisco that reported a 31% increase in the odds of unsuppressed viral loads. This is despite retention in care recorded via telemedicine [10].(d)Health workforce: This considers some challenges during COVID-19, such as low staffing levels (mainly nurses), unequal geographical distribution, and inadequate personal protective equipment (PPE) [2].(e)Medical products and technologies: this is related to PPE (masks, gloves, face shields, and gowns). It was affected during the pandemic due to the competition between countries or the overreliance on a few countries for production [2].(f)Finally, it includes the public health functions considering testing, contact tracing, quarantine or self-isolation, and surveillance to break chain transmission [2].

This study’s objective is to explore the impact of the COVID-19 pandemic on the health care services for PLWHA in Cochabamba, Bolivia. The generated information will allow us to strengthen the resilience of HIV health services and thereby decrease HIV transmission and reduce morbidity and mortality among PLWHA in future health crises.

## 2. Materials and Methods

### 2.1. Research Team and Reflexivity

The interviews were conducted by the principal investigator (LA), a medical doctor and a doctoral candidate in public health. She has worked in various consultancies related mainly to the issue of HIV/AIDS and has training and experience in qualitative studies and interview development.

Before the start of the study, the researchers had no direct relationship with the participants. Participants were invited by explaining the objectives of the study and the researcher’s interests.

### 2.2. Study Design

This qualitative and observational study was implemented in health care services for PLWHA in Cochabamba, Bolivia between January and March 2022. Since there is very little information on the impact of the COVID-19 pandemic on the operation of the centers in Bolivia, we used a qualitative approach to explore this, eliciting the perceptions and experiences of the health personnel. For this reason, in addition to the small number of health personnel working in the PLWHA care centers, interviews were preferred over surveys. An ethnographic method was used within the interpretive paradigm to carry out the thematic analysis since phenomena are explored in light of the social, cultural, political, and physical environments surrounding the people they are studying, considering a holistic approach [21].

### 2.3. Population and Sampling Strategy

The study considered centers that provide care to people with HIV and “differentiated provision of services” as sampling units. These centers offer comprehensive services for prevention, diagnosis, treatment, and support of HIV infection. Additionally, they are responsible for initiating and maintaining ART, the monitoring of response, providing support for treatment adherence, switching to second-line and third-line ART, and providing palliative care if needed. The National public health care system treats opportunistic diseases associated with HIV/AIDS with less complex issues being treated at the Primary Care Centers, and more complex problems at the second and third levels. The treating physicians of PLWHA usually coordinate these referrals. According to national regulations, these centers are to provide services to meet the needs of people with HIV infection and thereby to reduce the burden on the health care system. This public health approach aims to ensure the broadest possible access to high-quality services at the population level [8]. However, these centers have minimum staffing for their operation (doctor, consultant, laboratorian, pharmacy manager, and support staff in nursing and statistical information).

We used an intentional sampling strategy, including key informants working in each institution attending PLWHA (HIV/AIDS care centers). Health staff (doctors, nurses, and psychologists) who could provide information about the development, implementation, or maintenance of PLWHA care and treatment were considered key informants. The selection of key informants was to represent the relevant institutions.

Until 2020, four types of centers were operating in Cochabamba that provided differentiated care, from which the study sample was obtained (Table 1):(a)Departmental Surveillance, Information, and Referral Centers for HIV/AIDS (CDVIR, Cochabamba): The CDVIR in Cochabamba is a first-level care center located in the capital city to provide comprehensive care. It is technically dependent on the Departmental Health Services, Ministry of Health (SEDES, Cochabamba) and functions as the operational arm of the Departmental STD/HIV/AIDS program.

From this center, three key informants were invited to participate in the study: the head physician, the head of nursing, and the center’s psychologist.

(b)Centers for differentiated care (four centers; two urban and two rural): They are public dependency centers in health facilities that have trained personnel for comprehensive care of PLWHA.

The heads of each center were invited to participate in the study. The interview with the director of one of the centers in urban areas could not be arranged due to work overload and limited time availability. Therefore, information could only be collected from three of the four operating centers.

(c)Decentralized center (one center) This was a new public dependency center located in a rural area. It provides antiretroviral treatment and helped improve the dispensing of ARV drugs and the follow-up of PLWHA.

The head of the decentralized center was invited to participate in the study.

(d)Private centers: In Cochabamba, only one private health center was identified (depending on an NGO), which provided comprehensive care to PLWHA. This center is located in the capital city (urban area).

Two professionals were invited to participate in the study: the NGO director and the medical director.

Additionally, the departmental head of the STI/HIV/AIDS program, dependent on the Ministry of Health (SEDES, Cochabamba, Bolivia), was invited to participate.

### 2.4. Data Collection

We conducted semi-structured individual interviews with ten health professionals. Before the interview, the participants answered a short sociodemographic and work questionnaire considering: (i) gender, (ii) profession, (iii) type of service in which they work, (iv) area of work (management and provision of services), and (v) subsector (public, private for-profit, private non-profit). For the in-depth interviews, we applied a narrative methodology. It allowed for the introduction of a topic, and for proposing some hypothetical situations to address some issues related to the research problem [21]. An interview guide was developed considering the determinants of the health systems resilience framework set by the WHO. The interview guide included open-ended questions considering six a priori dimensions: the first dimension, related to governance, finance, and collaboration across sectors, was studied through three categories: (i) provision of health coverage, (ii) financing, and (iii) changes promoted in the institutional structures. The second dimension, health service delivery, was characterized in terms of three categories: (i) healthcare provided and organizational changes within PLWHA communities, (ii) provision of medications, and (iii) laboratories. The third dimension, the health workforce, considered two categories (i) changes in human resources and (ii) the effect on the health personnel. The fourth dimension, medical products and technologies considered the provision of PPE. The fifth dimension refers to community engagement during the pandemic, exploring the participation of collective PLWHA. The last dimension referred to functions of public health considering promotion and prevention activities.

The interviews lasted 15–45 min. Eight interviews were conducted face to face at the respective health services, and two were held virtually using Zoom.

### 2.5. Data Analysis

The recordings of the interviews were transcribed verbatim and uploaded to the Atlas.ti program. We used an inductive thematic analysis approach, where researchers initially read the transcripts to identify primary and minor nodes and developed a final set of “core” codes. Two trained researchers with divergent fields of expertise participated in the analysis.

### 2.6. Ethical Considerations

The study protocol was approved by the ethics committee of the Faculty of Health Sciences of the Universidad del Valle (UNIVALLE) in Cochabamba, Bolivia. International ethical considerations established in the Declaration of Helsinki were included in all the research steps. The subjects’ participation was voluntary and verbal consent was obtained from the participants before participating in the interviews. We kept their identification and the institutions for which they reported confidential.

## 3. Results

A total of ten key personnel from six centers attending PLWHA participated in the study. The person in charge of one of the centers could not be contacted due to work overload and little time availability. Most of the participants were male (60%), medical doctors (60%), working in urban areas (80%), and in charge of administrative tasks (70%). The average time working with PLWHA was ten years (Table 2).

The dimensions initially proposed for the data collection were maintained in the analysis. A summary of all themes and subthemes is shown in Figure 1 and described below.

### 3.1. Governance, Finance, and Collaboration across Sectors

The information analyzed shows how health centers for PLWHA had to face the impact of a pandemic on an unexpected scale. The COVID-19 containment and mitigation measures implemented by the central government also affected the regular care and provision of services for people with PLWHA. It included a period of strict lockdown (three months) that gradually became more flexible. Thus, healthcare also prioritized the care of COVID-19 patients, postponing or adapting the care of other pathologies. Although the WHO framework also considers the regulatory measures for this dimension, the availability of specific COVID-19 guidelines for PLWHA was not mentioned. Three categories emerged in this dimension: (i) provision of health coverage; (ii) financing, and (iii) changes promoted in the institutional structures, which are described below.

a.Provision of health coverage

Comprehensive care for PLWHAs was limited due to the health system’s capacity, but it was not suspended. Monitoring and controls have been carried out to guarantee attention.


*“We have gone to monitor, do the supervision, and we have seen that there were no patients, but the staff was ready to attend, they have not suspended, it has only been from the external part, if it is worth the term of the external clients who have not come to services for fear of contagion”*
(Interviewee-1).

The coordination and support of international networks was highlighted to guarantee emergency expenses and continue with care, complying with the recommended biosafety measures.


*“...to face of this situation at the international level of these two networks immediately sent us funds to cover these urgent expenses: food and biosafety, alcohol, masks, and gloves for the health service, etc.”*
(Interviewee-2).

Difficulties were also expressed between national and international recommendations, especially at the beginning of the pandemic, when there was a lot of uncertainty in patient care protocols.


*“(decision making) It has been very difficult because the public health system has managed its own regulations, they have relied on the regulations that they have, and we have relied more on the international guidelines of our cooperators”*
(Interviewee-3).

b.Financing

The funding for the operation of the HIV/AIDS program was maintained during the pandemic. The relevance of international support was recognized, especially in facing the pandemic in these vulnerable groups. All the central authorities of care for PLWHA highlighted the support of international partners who helped provide personal protective equipment and food for PLWHA. However, unlike a center dependent on an NGO, this was perceived as a late response.


*“Until now, emergency funds are being distributed, which is no longer an emergency”*
(Interviewee-2)

On the other hand, one interviewee expressed concern about the sustainability of laboratory reagents.

*“The problem of the laboratories definitely… I do not think it is right continue subsidizing, I think, because the Global Fund is decreasing, and they are closing”*.(Interviewee-5)

c.Changes promoted in the institutional structures

The interviewers reported that all the health services that attend PLWHA in urban areas adapted their infrastructures to protect the safety of the patients.


*“Considering that these patients suffer from immunosuppression, so they try to avoid contact with other patients”*
(Interviewee-8).

Also, the rural and main center’s infrastructure was recognized as a limitation since the physical space was not enough to provide good care, and taking care of confidentiality.


*“The space is too small to attend many people; possibly we are walking sideways, I mean, there is no space”*
(Interviewee-10).


*“The infrastructure did not increase; to tell you the truth we didn’t even have space for Covid, so it was more difficult to have a specific area for HIV. We do not have the infrastructure for them”*
(Interviewee-9).

Unfortunately, in rural areas, adding to infrastructure problems, biosecurity was not possible due to the population’s resistance, exposing health personnel. It was a concern as the physical space was not suitable for the recommended physical distancing.


*“There are some rebels with the use of biosecurity measures. Many people here have normally been working. The patients did not isolate themselves in their homes. They did not use masks that have generated discomfort between the doctors”*
(Interviewee-9).

### 3.2. Health Service Delivery

The centers that serve PLWHA also had to adapt the provision of services according to the requirements at the national level. It included the modification of work schedules and shifts of health personnel, limitations in the physical evaluation of patients, and the implementation of new strategies to maintain the provision of medications. Thus, difficulties were also expressed between the different levels of care of the public health system since the centers that serve PLWHA were not enabled to attend to cases of COVID-19. This situation forced the care of PLWHA patients in other centers with little experience in this kind of care.

In general, the demand for patient care was reduced due to fear (COVID-19 transmission), limitations in mobility (access to transportation during lockdown), or changes in the provision of services. This situation is expressed with concern for the medium or long-term consequences in relation to the evolution of already diagnosed patients and also for the delay in the diagnosis of new patients.

On the other hand, the pandemic also offered new opportunities for the modernization and use of technology in patient follow-up. Although some sectors valued this, this was very inequitable between urban and rural areas, to the detriment of the latter due to limited resources and internet access.

For this dimension, three categories were analyzed: (i) healthcare provided and organizational changes within PLWHA communities, (ii) provision of medications, and (iii) laboratories.

a.Healthcare provided, and organizational changes within PLWHA communities

Although the centers were unprepared to face the emergency, the provision of health services was maintained. However, at the peak of the waves of the pandemic, attention was prioritized to critical situations, and regular health checks or care for non-critical patients were suspended.


*“COVID caught us on a curve. It is the absolute truth, as a health institution, we have practically stopped many programs prioritizing the attention to COVID-19, without neglecting our functions”*
(Interviewee-1).

The provision of health services was continuously face-to-face, especially in rural areas, where services were not suspended. In urban areas, the interviewees reported the adoption and scaling up of digital technology or telehealth services as a strategy to reduce this distance, generating a decrease in contact with people with HIV/AIDS. This condition has modified traditional healthcare facilities and providers, an important aspect that affected the link between health personnel and PLWHA. Interviewees from urban areas expressed difficulties in physical contact.


*“Practically, these patients have not been checked, as we used to do”*
(Interviewee-4).

Contradictorily, this situation was positively valued by other interviewees, especially among psychologists, since it allows for maintaining confidentiality and trust with the patient.


*“We have had more intimate contact, the possibility that calls or talk by phone gave us more confidentiality; because before the pandemic, the role of the psychologist, dentist, doctor or the patient was well defined, in other words, they had a hierarchical dialogue, so the calls allowed more contact... kind of trustworthy; so this helped us to understand each other”*
(Interviewee-6).

On the other hand, the centers implemented different strategies to reduce working hours/days. In the complex hospitals, they developed alternating shifts to avoid massive infections.


*“We have changed the schedules. Before they were discontinuous schedules, we attended all day, but also due to the issue of transport, of the users who could not arrive, a running schedule has been made, let’s say where the PLWHA have been attended, but now in another context”*
(Interviewee-1).

The differentiated care centers for PLWHA care did not attend cases of COVID-19 infection, suspected cases or COVID-19 positive patients. For that, PLWHA were referred to other reference centers of the public system, where treatment for COVID-19 was provided. In this process, some difficulties were reported in the referral and counter-referral process since, in many care centers, there was no predisposition to attend PLWHA.


*“They reject our patients, we had patients for surgery, but they did not do surgeries. The patient has difficulty in that part; they do not take care of … the patient’s health, they reject them”*
(Interviewee-7).


*“We refer patients to the third level, to the infectious disease sector or other services, depending on the patient and obviously these hospitals have not been working regularly, so they were patients difficult to attend, especially those complicated or with the opportunistic disease, so we have had to adapt to the situations”*
(Interviewee-4).

b.Provision of medications

The interviewers reported that the distribution of medicines to PLWHA was fundamental and a priority among different authorities. They used various strategies such as house-to-house distribution, guaranteeing confidentiality.


*“Keeping the confidentiality, we put in a package… I remember I bought a couple of boxes of chocolates, and I put the medicines in the middle in order to guarantee confidentiality”*
(Interviewee-1).

Additionally, measures such as the delivery of ARVs for more time and the delivery of ARVs to family members were assumed as exceptional measures.


*“Another right measure was not to give only medication for one month otherwise for three months, two months; they were given to the patients for more time”*
(Interviewee-12).


*“All has been coordinated, so if the patient was unwell, or was isolated due to another illness, as a cold or a fracture, in that case the family member could picked up the medication for him”*
(Interviewee-7).

Although the efforts to deliver medications by health personnel on time were expressed, most interviewees mentioned disrupting access to ARV treatments, especially in rural areas


*“Patients were being left without medicines”*
(Interviewee-10).

These difficulties may have led to a progression of the infection to the AIDS phase,


*“We have not had cases in the AIDS phase, there were none, but after the pandemic, they have appeared”*
(Interviewee-7).

c.Laboratories

The complementary laboratory tests for PLWHA were limited during the pandemic because the medical staff in charge of these tests got COVID-19 and also the lack of reagents. It generated gaps and delays in laboratory controls, affecting urban and rural areas.


*“Access, transportation, programming failed; I mean the chain of supplies for the laboratory has been interrupted in some moment”*
(Interviewee-3).

These difficulties in carrying out laboratory tests have caused longer waiting lists for results, and decreases in the identification of new cases and late diagnoses, which could have an impact in the medium and long term.


*“We are not sending patients, because we have five spaces to send according to the schedule, right now it is busy until April”*
(Interviewee-5).


*“We used to diagnose one new case per day, and at the time of the pandemic it has reduced to one per week, sometimes even every two weeks, diagnoses were not frequent at the time of the pandemic”*
(Interviewee-6).

### 3.3. Health Workforce

The interviewees mentioned the impact that the COVID-19 pandemic had on health personnel. In many countries, COVID-19 has spread quickly among health workers. They have also been the most exposed to the virus [2]. In Bolivia, the health staff that attends to HIV patients reported limited health personnel availability at health care centers, which exacerbated a limitation before the pandemic, especially in rural areas. Health personnel also reported exhaustion and fear, especially in rural areas where the population did not comply with the biosafety protocols established at the national level (e.g., use of masks and physical distancing). This dimension considered of two categories: (i) changes in human resources and (ii) the effect on health personnel

a.Changes in human resources

One of the interviewees explained that the reduction in health personnel was mainly due to COVID-19 infection, preventive isolation, licenses for being part of a risk population, the limitation for mobilization and the reorganization of services (e.g., absence of medical students in hospitals):


*“With the presence of COVID, many people have panicked in health services, as a result, many health personnel has taken their vacations or their permissions; using the different regulations in order to protect their health, also we saw that many human resources have forced their retirement because they felt panic”*
(Interviewee-1).

In addition, the geographical distribution of personnel is perceived as inequitable, being less in the rural areas, reflecting the internal conflict between physicians, who perceived the number of professionals as insufficient.

b.Effect on the health personnel

The interviewees recognized some situations that affected health personnel. On the one hand, the rotation of health personnel affected the regular functioning of two health services. On the other hand, the limited staffing was exacerbated by the pandemic, especially in rural centers, where the situation was already challenging.

Even with these measures, most interviewees who work in care felt tired and still perceived work overload, especially in public services.


*“We need more human resources; sometimes we are not enough”*
(Interviewee-5).


*“… with the presence of COVID, many people have panicked in health services. As a result, many health personnel has exhausted all their vacations, or their permits, to the extreme. They have taken advantage of the different provisions that were in place to protect also the health of our health personnel. Still, we have seen that many human resources have forced their retirement because there was direct panic”*
(Interviewee-1).

Patients from rural areas did not get used to biosafety measures, exposing health personnel and generating panic among them.


*“There are some rebels with the use of biosecurity measures. Many people here have normally been working. The patients did not isolate themselves in their homes. They did not use masks that have generated discomfort between the doctors”*
(Interviewee-9).

### 3.4. Medical Products and Technologies

Like other countries, the health centers in Bolivia presented difficulties in providing personal protective equipment, especially for health personnel at the beginning of the pandemic. In Bolivia, a strict lockdown was established during the first months of the pandemic that allowed, in a way, to have a time window for the organization of services and the provision of the necessary equipment. Thus, at the national level, free vaccination of the population was guaranteed, prioritizing risk groups initially, including PLWHA. This situation is reflected in this dimension since a few problems were reported at the time of the interview.

For the provision of medical supplies, international collaboration and work of the NGO are also highlighted to guarantee this equipment to the PLWHA.


*“The National Program has provided personal protection equipment to people living with HIV, they gave masks, alcohol sprays, and we have given the leaders a biosafety suit and masks”*
(Interviewee-1).

The access and use of technology were presented unequally between urban and rural areas and among the people who could access it. It also affected access to health care, regular check-ups, and other information provided during this period.


*“… there are patients who do not have access to the WhatsApp either, so that is the problem, in patients who are from rural areas that are complicated, that they come here, right? We have given ourselves ways to attend, but I believe that the attention to these patients has been affected”*
(Interviewee-4).

### 3.5. Community Engagement

Engagement with local communities is central to resilient health systems as a way to inform service delivery, decision-making, and promote governance. It helps meet communities’ needs before, during, and after crises [2]. In Cochabamba, the work with PLWHA organizations has been reported to be a fundamental piece for coordinating different strategies since before the pandemic. During the health crisis, this collaborative work has been maintained for specific actions such as the delivery of medicines and the referral of patients to levels of greater complexity in the public health system. This coordination has also been done through informal media such as WhatsApp to respond quickly to needs.


*“There has been more direct coordination with them; sometimes we have met, other times we have contacted through social networks, through the WhatsApp that has been key to organizing ourselves in a better way. For example with the Mobile Unit, it had an area where we had to go to leave the medicines, for people living with HIV and we have accompanied them in these processes as well”*
(Interviewed-1).

An interviewee expresses support for the patient referral process in the following quote.


*“He helps, with other levels in the hospital, for example … where reject us, patients”*
(Interviewed-7).

Work with “pairs” is also highlighted to achieve better results in treatment during this period.


*“Peers are the ones who collaborate the most so that antiretroviral therapy is effective in these patients”*
(Interviewed-10).

On the other hand, due to the mitigation and containment measures, many PLWHA lost their jobs, reduced their economic income, or had to be quarantined. In these situations, different actions were coordinated, such as distributing food to those most affected. Additionally, PLWHA organizations supported national strategies, mainly referring to delivering Personal Protection Equipment (face masks) and applying COVID-19 vaccines.

However, although there have been no specific prevention campaigns for COVID-19 for PLWHA to explain and motivate compliance with the recommendations, one interviewee highlighted the adherence of PLWHA to biosecurity measures.


*“In brackets… I would like to tell you that people living with HIV have fully complied with all biosafety measures, and this must be highlighted. Many patients have sheltered in their homes and have not left their homes because they... the WHO has qualified that HIV/AIDS has qualified as an underlying disease”*
(Interviewed-1).

### 3.6. Functions of Public Health

In Bolivia, the management of the pandemic was initially centralized for the diagnosis and implementation of mitigation and containment measures. Subsequently, in each region of the country, capacities and infrastructure for diagnosis and regulations were strengthened according to the local context. In that sense, some health centers were adapted for the exclusive attention of suspected patients or positive cases of COVID-19. The institutions in charge of caring for patients with HIV did not carry out tests or isolation of patients with COVID-19. This led to the lack of coordination of some actions and the need for the intervention of PLWHA organizations to guarantee the respective care of referred patients. Some coordinated efforts are mentioned to support patients during quarantine periods and care for the target population. However, specific strategies to support the diagnosis and traceability of PLWHA people did not appear in the interviews.


*“We coordinated with the CDVIR, and since there were rigid quarantines, it was impossible for people to come, go to our health services, so work groups have been formed to take the medicines to their homes”*
(Interviewee 1).

*“They were very affected and had nowhere to turn then, and we were there at the right time and we coordinated with sexual diversity, sex workers, and people with HIV through our leadership and the presence of IDH in these populations and especially with our patients. We have an active queue of 280 patients; Since some of them are extremely poor, everything was solved virtually, so the key was to work with the population, their leaders so that they help us distribute all of this, identify those who are most affected”*.(Interviewee 2)

Likewise, many prevention and health promotion campaigns focused on the general population and COVID-19, neglecting other health problems. Most interviewees report a reduction in actions to attract new patients, reducing access to prevention and the early diagnosis of HIV. HIV/AIDS promotion and prevention strategies, such as health education, public campaigns, and condom distribution, were suspended. This situation is shown with concern because it could affect very young populations and sexual workers:


*“They get pregnant very young and is where they have been acquiring HIV, from 15 to 20 years old it is a very vulnerable group for us”*
(Interviewee 9).


*“All the impact of COVID has even delayed the diagnosis, that is, now we are beginning to diagnose what happened a long time ago, because sex workers, despite COVID, have continued to work after the quarantine, or during the quarantine”*
(Interviewee 2).

Nevertheless, supplementary breastfeeding prevention of mother-to-child transmission ART was maintained, as expressed by one of the interviewees.


*“The impact of COVID has delayed the new diagnosis, which means that we are beginning to diagnose what happened long ago”*
(Interviewee-2).

## 4. Discussion

The COVID-19 pandemic has caused unprecedented global health, economic, and social impacts [22]. The present qualitative study was carried out in Cochabamba-Bolivia, including the points of view of the key health staff. It helps us to understand how the management of HIV/AIDS health services was during the pandemic. Their perspectives provide operational strategies to address the resilience of health services.

The pandemic has affected different aspects of health services, including control access, ARV treatments among PLWHA, HIV prevention, and early diagnosis, especially among risk groups. The lessons learned must guide the strategies to reduce the impact of the pandemic [16]. It is consistent with other studies. In Guatemala, the diagnostic services for HIV have been severely affected and have increased the deaths from opportunistic infections [23]. Ethiopia also reported a significant decrease in HIV testing and detection [12].

The pandemic has had an impact on all the dimensions studied. Concerning the governance and financing dimension stands out the creation of a new decentralized center in the rural area due to the demand of the patients, but also conflicts between decentralized health centers and central authorities. These difficulties were also present in other countries. In South Africa, little collaboration among healthcare providers was reported to address Type 2 Diabetes and HIV/AIDS comorbidities. It was mainly due to poor communication, noncentralized patient information, and staff shortages [24]. These deficiencies in coordination and the interruption of services foresees a worrying scenario for access to health services.

Regarding the health service delivery dimension, although they had to adapt the infrastructure to continue PLWHA care, the main problem reported was the size of the centers. Communication technologies were fundamental during the pandemic, becoming an essential tool in the care of PLWHA, especially in urban areas. It was similar to what was reported in Southern California, with the use of telehealth in delivering medical and non-medical HIV care services. The principal barriers identified were technological challenges, digital literacy, client/provider experiences, the low socio-economic status of the client population, and reimbursement issues [25]. This situation could be similar in Cochabamba, recognizing opportunities for future improvement. In the healthcare system, it is important to consider strategies to overcome barriers to the use of technology, including regulations and legal aspects [26]. In this sense, in our study, psychologists better valued the incorporation of new technology in the care process than medical doctors. Psychologists mentioned that reliability and confidence in communication with patients increased. It is consistent with other studies, where the implementation of telemedicine, especially in children and young people, has had good results in addressing mental health problems [27]. On the other hand, although the doctors adopted technology in our study, they prefer face-to-face care. However, previous experiences have been reported on the use of telehealth, where the emphasis is placed on effective communication with PLWHA that empower users and health services, especially favoring the most vulnerable groups [28].

As reported in other countries, the care demand for PLWHA decreased at the beginning of the pandemic. One study in Nigeria showed that patients could not access services because they could not leave their houses during the lockdown (31.91%) of the participants, or there was no transportation (18.13%) [29]. In that sense, some drug delivery strategies were adopted, such as house-to-house distribution responding to the population with care needs [3]. In the case of PLWHA and minimizing the adherence problems, a study in four countries suggests that the home delivery model is client-centered, improving the quality of life for PLWHA by providing a convenient means of uninterrupted access to ART [30].

The situation of health workforce dimension also presented difficulties. On the one hand, the health personnel was more exposed to COVID-19 contagion and death; on the other hand, the pandemic produced exhaustion due to the intensity of work among personnel. Several countries reported psychological support for health workers, such as counseling or trauma support, to maintain well-being [2]. But it was not the case for the personnel who attend PLWHA in Cochabamba. In situations of a health crisis, such as COVID-19, it is essential to consider support measures for health personnel. It has been reported that encouraging work in well-integrated teams and perceived support from hospital management can mitigate stress and care for health personnel [31].

Regarding the medical products and technologies dimension, our study did not report problems in the provision of PPE, unlike in Kenya, which was affected by the first wave of the lack of personal protective equipment [32].

Finally, concerning the community engagement dimension, we found the active participation of the PLWHA organizations in specific actions, especially in urban areas. It is consistent with other studies where communities and patients are vital contributors to systems improvement through program integrations and community participation [33].

In general, our study found no relevant differences in the interviewees’ discourse according to gender, profession, location (urban/rural), or years of experience. In the central themes, the speeches were consistent, finding information saturation for some themes.

Our study has some limitations. Due to the time limitation of one director, it was impossible to include information from one of the centers for differentiated care in urban areas. Although the information may be somewhat similar to the other centers included, it is possible that it had some particular characteristics that could not be explored. It is possible that other barriers and challenges that present mainly in urban areas would have emerged. During the study, the director of that center did not have time availability due to the new demands and challenges generated by the COVID-19 pandemic, both in health centers and at the level of professionals who provide medical care.

Finally, although all potential safeguards were taken during the interviews, it is possible that some participants felt uncomfortable discussing or sharing sensitive information about their jobs, representing a study limitation.

## 5. Conclusions

The COVID-19 pandemic has affected the prevention, diagnosis, and adherence to ART, especially in high-risk groups. To reduce the pandemic’s impact, it is crucial to work on the resilience of care services attending PLWHA. For that, we need a more holistic view, as proposed by the dimensions of the determinant of health services framework, to address PLWHA in Cochabamba. The implementation of integrated care considering alternative strategies like telemedicine, self-sampling, and self-testing technologies have been used before with promising results. It showed a potential to scale up, overcoming human and financial limitations, especially in health emergency contexts.

## Figures and Tables

**Figure 1 ijerph-19-13515-f001:**
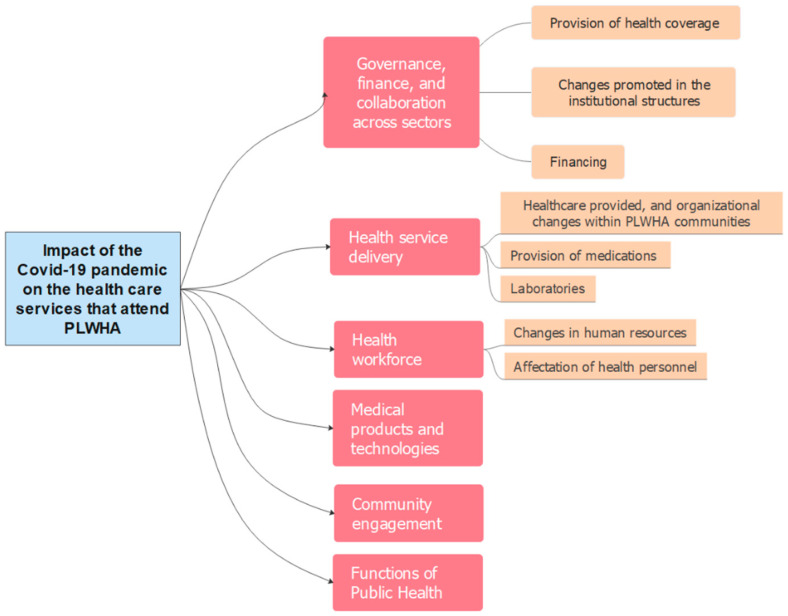
Coding tree identification regarding the impact of the COVID-19 pandemic on the health care services that attend PLWHA.

**Table 1 ijerph-19-13515-t001:** Characteristics of Health Services that attend PLWHA in this study.

	Type	Place	Number of Centers	N° of Health Workers (Total)	N° of Interviewees
a	Departmental Surveillance, Information and Referral Centers for HIV/AIDS (CDVIR)	Urban	1	7	3
b	Centers for differentiated care	Urban	2	2	2
		Rural	2	2	1
c	Decentralized center	Rural	1	1	1
d	No Governmental Organization	Urban	1	5	2
e	STI departmental program	Urban	1	1	1

Source: Own elaboration based on interviews.

**Table 2 ijerph-19-13515-t002:** Participants’ type of organization and years of experience working with people living with HIV/AIDS (PLWHA).

Number of Interviews	Sex	Area	Profession	Area of work	Years of Experience Working with PLWHA
1	Male	Urban	Psychologist	Administrative	20 years
2	Male	Urban	Medical Doctor	Administrative	25 years
3	Male	Urban	Medical Doctor	Clinical	15 years
4	Male	Urban	Medical Doctor	Administrative	14 years
5	Female	Urban	Nurse	Administrative/care	4 years
6	Male	Urban	Psychologist	Clinical	8 years
7	Female	Urban	Nurse	Administrative/care	5 years
8	Male	Urban	Medical Doctor	Clinical	3 months
9	Female	Rural	Medical Doctor	Administrative/care	5 years
10	Female	Rural	Medical Doctor	Administrative/care	2 years

## Data Availability

The datasets generated during and/or analyzed during the current study are available from the corresponding author upon reasonable request.

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
