# Peer review of "Response of Care Services for Patients with HIV/AIDS during a Pandemic: Perspectives of Health Staff in Bolivia"

_ijerph, 2022, doi:10.3390/ijerph192013515_

Round 1
Reviewer 1 Report (Previous Reviewer 2)
I thank to the authors for effort on the changes and improvements made on the manuscript.
I congratulate you on the subject on which you focus your study.
Author Response
Dear Editor:
Thank you very much for reviewing our manuscript entitled “Response of care services for patients with HIV/AIDS in times of pandemic: perspectives of health staff in Bolivia” and for giving us the opportunity to improve it and submit a new revised version. The comments have helped us review the results again, clarify some points, complement the missing information, and organize our results better. We are very grateful for this process.
In the revised manuscript, you can find the modifications of the text in “track changes mode”. We consider that the previous reviewers' recommendations have significantly improved our manuscript.
Reviewer 1
I thank to the authors for effort on the changes and improvements made on the manuscript.
I congratulate you on the subject on which you focus your study.
Author Response:
Thank you very much for the comment. It encourages us to continue researching this very relevant topic. We also appreciate previous comments and suggestions that helped us improve the manuscript.
Reviewer 2 Report (New Reviewer)
I really appreciate your work, for scientific and ethical reasons. It is really interesting to know what happened during the Covid-19 crisis, not only in Western countries, but also in the rest (most) of the world.
For these reasons, I must suggest that you provide adequate bibliographic support for your work. It is especially important, in a medical context, to provide a clear scientific framework of the methodology used. In several lines of work, I suggest where this is strictly necessary. For instance, it is important to refer to the narrative approach, or explain what semi-structured interviews are, and so on.
Author Response
To: IJERPH Editorial Office
Re: Manuscript ID: ijerph-1827859
Type of manuscript: Article
Title: Response of care services for patients with HIV/AIDS in times of pandemic: perspectives of health staff in Bolivia
Dear Editor:
Thank you very much for reviewing our manuscript entitled “Response of care services for patients with HIV/AIDS in times of pandemic: perspectives of health staff in Bolivia” and for giving us the opportunity to improve it and submit a new revised version. The comments have helped us review the results again, clarify some points, complement the missing information, and organize our results better. We are very grateful for this process.
In the revised manuscript, you can find the modifications of the text in “track changes mode”. We consider that the previous reviewers' recommendations have significantly improved our manuscript.
Below we answer (blue color) each of the observations made:
Reviewer 2
I really appreciate your work, for scientific and ethical reasons. It is really interesting to know what happened during the Covid-19 crisis, not only in Western countries, but also in the rest (most) of the world.
For these reasons, I must suggest that you provide adequate bibliographic support for your work. It is especially important, in a medical context, to provide a clear scientific framework of the methodology used. In several lines of work, I suggest where this is strictly necessary. For instance, it is important to refer to the narrative approach, or explain what semi-structured interviews are, and so on.
Thank you very much for the comment and suggestions to improve the work. We agree with you on the importance of bibliographic references and adequate explanation and foundation in different parts of the research process. In this new version, we have incorporated the bibliographic reference of some additional studies that strengthen the work:
Cunha, G.H.d.; Lima, M.A.C.; Siqueira, L.R.; Fontenele, M.S.M.; Ramalho, A.K.L.; Almeida, P.C.d. Lifestyle and adherence to antiretrovirals in people with HIV in the COVID-19 pandemic. Revista Brasileira de Enfermagem 2022, 75.
Queiroz, A.A.F.L.; Mendes, I.A.C.; Dias, S. Barreiras de acesso à profilaxia pós-exposição ao HIV: estudo de caso. Acta Paulista de Enfermagem 2022, 35.
Angel, J.C.A.; Martínez-Buitrago, E.; Posada-Vergara, M.P. COVID-19 and HIV. Colombia Médica 2020, 51.
Thus, we have also better explained the definition and bibliographic support of the method and techniques used in the investigation based on this author:
Ulin, P.R.; Robinson, E.T.; Tolley, E.E. Investigación aplicada en salud pública: métodos cualitativos; Organización Panamericana de la Salud: 2005.
This manuscript is a resubmission of an earlier submission. The following is a list of the peer review reports and author responses from that submission.
Round 1
Reviewer 1 Report
In general, I found this to be an interesting report about an important problem. The authors do a good job in describing the problem of the impact of the pandemic on the care of patients with HIV.
The authors correctly state that this is a qualitative study, but I feel that it is also observational because the authors are systematically studying the statement from the interviews. There is no mention of why this method was selected. A statement as to why they selected interviews as opposed to a survey would be useful.
The authors do not sufficiently describe how the 10 interviewees were selected. There is no mention of random selection or volunteers. The do describe the experience level, gender and whether they work in a rural or urban area.
How many facilities did the authors have to select from?
How do the authors define "key personnel?"
The authors state "one of the centers could not be contacted." Why, was it in a rural area? Did attempts to contact fail? How is the reader to understand how this lack of communication impacted the study?
Author Response
- Response to comments by Referee 1
In general, I found this to be an interesting report about an important problem. The authors do a good job in describing the problem of the impact of the pandemic on the care of patients with HIV.
Thank you very much for your comments
The authors correctly state that this is a qualitative study, but I feel that it is also observational because the authors are systematically studying the statement from the interviews. There is no mention of why this method was selected. A statement as to why they selected interviews as opposed to a survey would be useful.
Thank you very much for your observation and suggestion. Given the study's objective, it has been preferred to work with a qualitative approach through interviews instead of a quantitative approach. We are interested in analyzing the experiences and perceptions of a phenomenon in depth and little studied.
It has been clarified in the new version of the manuscript. We added that it is an observational study. In addition, we have justified the choice of the qualitative approach and our selection of interviews instead of surveys.
Line 135-141:
“A qualitative and observational study was implemented between January and March 2022 in health care services that attend PLWHA in Cochabamba, Bolivia. Since there is very little information on the impact of the Covid-19 pandemic in these centers in Bolivia, we used a qualitative approach to explore this phenomenon in greater depth and explore the perceptions and experiences of health personnel. For this reason, in addition to the small number of health personnel working in the PLWHA care centers, interviews were preferred over surveys”
The authors do not sufficiently describe how the 10 interviewees were selected. There is no mention of random selection or volunteers. The do describe the experience level, gender and whether they work in a rural or urban area.
Thank you very much for this observation. We have realized that the context and population description were unclear and incomplete and did not allow us to understand the sample selection process properly.
In this new version, we have reformulated the full subtitle: "Population and sampling strategy"- lines 143-194
How many facilities did the authors have to select from?
Thank you for pointing out this. In the revised version, we have clarified this point in relation to the previous point.
How do the authors define "key personnel?"
Thank you very much for your important observation. In the revised version, we have clarified that the health staff working in the health centers for the care of PLWHA (HIV/AIDS care centers) who could provide information about the development, implementation, or maintenance of PLWHA were considered key informants.
Line 161-164
“We used an intentional sampling strategy, including key informants working in institutions attending PLWHA (HIV/AIDS care centers). Health staff (doctors, nurses, and psychologists) who could provide information about the development, implementation, or maintenance of PLWHA care and treatment were considered key informants. This selection considered the institution where health staff worked, not the person”
The authors state "one of the centers could not be contacted." Why, was it in a rural area? Did attempts to contact fail? How is the reader to understand how this lack of communication impacted the study?
Thank you for highlighting this point. Despite multiple attempts, the interview with the director of one of the centers in the urban area (Deconcentrated Centers) could not be arranged, due to work overload and limited time availability.
Line 178-181
The heads of each center were invited to participate in the study. The interview with the director of one of the centers in urban areas could not be arranged due to work overload and limited time availability. Therefore, information could only be collected from three of the four operating centers.
We have also incorporated this point into the discussion (Line 423-430).
Reviewer 2 Report
Dear Authors,
The topic is so important. This paper increases the readers ' awareness of the situation of HIV/AIDS in some countries, especially in Cochabamba (Bolivia). It is also described the effort made to manage this problem through the local health care workers during the Covid-19 pandemic situation.
But unfortunately, the application of qualitative design, sample, and especially reporting is weak.
I have the following major comments:
1. In the Introduction section, although the first paragraph gives the reader the idea that study takes place in South America, it has been compared with very different countries. Are the health care systems of those countries similar? What about employment conditions?
2. In the method section, the authors might need to justify and elaborate more the study design. I strongly recommend to use the COREQ guideline as guide to this study.
Unfortunately, the sample is not sufficient.
I see it is a qualitative study. But this does not mean it is possible to perform a study on such an important topic with 10 interviews. This is a major issue that makes the results insufficient.
Did you use any statements in the semi-structured form?
The authors can come up with suggestion about how they conducted the interviews; also, if the researchers received any experience or training before conducting the study. You should explain if the researchers have a relationship established prior to study commencement and also if many people refused to participate or dropped out; and explain the reasons.
3. In the results section, the scores were reported according to the different ages/ professions/ areas etc. but the authors did not discuss what is their significance or implications and what conclusion can the authors derive from analysing the scores according to these groups.
Major themes were described following WHO guidelines, but minor themes description was not reporting appropriately. It seems like themes were identified in advance or derived from the data. And sometimes I found that it was inconsistency between the data presented and the findings.
As pointed out in more detail in the major section of this review the manuscript should be improved in those areas.
Yours sincerely,
Author Response
Dear Authors,
The topic is so important. This paper increases the readers ' awareness of the situation of HIV/AIDS in some countries, especially in Cochabamba (Bolivia). It is also described the effort made to manage this problem through the local health care workers during the Covid-19 pandemic situation.
But unfortunately, the application of qualitative design, sample, and especially reporting is weak.
Thank you for your comments on the study's contribution and your observations regarding the design, sample, and report. Their assessments coincide with those of the other reviewer. We have become aware of some critical elements that we have not presented adequately, making it difficult to understand the study properly. In this new version, we have incorporated valuable observations, and we are sure that it has greatly helped to strengthen the report of the manuscript.
I have the following major comments:
In the Introduction section, although the first paragraph gives the reader the idea that study takes place in South America, it has been compared with very different countries. Are the health care systems of those countries similar? What about employment conditions?
We greatly appreciate your observation. We agree with you regarding including prior information on the impact on health services in Bolivia compared to other countries. Although Bolivia has some particular characteristics regarding the organization of the health system, it shares many elements with the health systems of other countries in Latin America. To address this observation, we incorporate in the introduction the reference and the results of a multicenter study (including Bolivia), where it is shown that in Bolivia the health system, and health teams, have been affected in the same way during the Covid - 19 pandemic. We believe it will help to contextualize the study better. (Lines 36-42)
On the other hand, and thanks to the observations of the two reviewers, we have better described the organization of the services that provide care to PLWHA, which is the main focus of the study.
In the method section, the authors might need to justify and elaborate more the study design. I strongly recommend to use the COREQ guideline as guide to this study.
We have now revised the document considering the COREQ guideline, and we have supplemented the information throughout the document.
Unfortunately, the sample is not sufficient.
Thanks for pointing out this. In the revised version, we have restructured the entire subheading "Population and sampling strategy" which was reported incompletely and unclearly. In the new version we explained that the study included six of the seven centers currently operating in Cochabamba that provide differentiated care for people living with HIV/AIDS. One of the centers could not be included due to work overload and little time available for its director.
Very few health personnel (doctors, nurses, and psychologists) work in each center. For that, we interview key informants who could provide information about the development, implementation, or maintenance of PLWHA care and treatment.
It is described now in lines 143-194
I see it is a qualitative study. But this does not mean it is possible to perform a study on such an important topic with 10 interviews. This is a major issue that makes the results insufficient.
We understand your concern regarding this point. It was our mistake not to have better explained the context, population, and sampling strategy in the previous version. To better explain this point, we have restructured the article by better describing these aspects, especially in the item "Population and sampling strategy". It is important to mention that we have included almost all the centers that serve PLWHA. On the other hand, during the pandemic, access to professionals in health centers was very limited.
Did you use any statements in the semi-structured form?
Yes. We used a semi-structured form for the interviews based on the WHO theoretical model for the resilience of health services. It considers five major dimensions with their respective categories described in lines 103-205
The authors can come up with suggestion about how they conducted the interviews; also, if the researchers received any experience or training before conducting the study. You should explain if the researchers have a relationship established prior to study commencement and also if many people refused to participate or dropped out; and explain the reasons.
Thank you very much for your observation. Also, as recommended in the COREQ guideline, we have clarified some elements concerning the conduct of the interviews.
The interviews were conducted by the first author (LA), who has training and experience in conducting interviews. The authors have previously worked on other studies related to HIV/AIDS, but there was no direct relationship established with the centers included in the study.
Before conducting the interview, the researchers contacted the participants to explain the objectives of the study and coordinate the interview process. After this, no person refused to participate; on the contrary, there was a lot of interest in describing the experience in their centers.
Only the director of one center could not be contacted because they had an overload of work and little availability of time. We understand that at the time of the interview, the center faced more urgent situations that required more attention.
These elements have been clarified in the material and methods section and the other elements suggested in the COREQ guideline.
In the results section, the scores were reported according to the different ages/ professions/ areas etc. but the authors did not discuss what is their significance or implications and what conclusion can the authors derive from analysing the scores according to these groups.
Thank you very much for highlighting this point. Indeed, table 1 shows the general characteristics of our interviewees and helped us to see some pattern in the discourse according to these characteristics.
In general, during the analysis, no relevant differences were found in the experience or perception of the interviewees according to these scores. It was only striking that professional psychologists perceived the online modality for patient care better than medical doctors. Psychologists even mentioned that reliability and confidence in communication with patients increased. At the same time, doctors, although they adapted to this modality, did not replace face-to-face medical attention in the same way.
In this revised version, we have incorporated these aspects into the discussion.
Major themes were described following WHO guidelines, but minor themes description was not reporting appropriately. It seems like themes were identified in advance or derived from the data. And sometimes I found that it was inconsistency between the data presented and the findings.
As pointed out in more detail in the major section of this review the manuscript should be improved in those areas.
Thank you very much for your critical comment and observation. For the analysis, we used an inductive thematic analysis approach. We initially read the transcripts to identify major and minor nodes and developed a final set of "core" codes. The dimensions and categories that emerged from the data coincided with the dimensions proposed for collecting information. Only one new category was included during the analysis (perceptions of repercussions on the care of PLWHA).
We realized that the organization of the dimensions and categories was a bit messy and some aspects had not been properly addressed in the presentation of results. In this new version, we have revised the description and consistency of the dimensions included.
All suggested corrections have been incorporated into the manuscript with the word's track change feature.
We consider that the recommendations of the reviewers have significantly improved our manuscript, and look forward to your response to the changes we have made.